# Pillar[6]arene acts as a biosensor for quantitative detection of a vitamin metabolite in crude biological samples

Masaya Ueno [1,2,10], Takuya Tomita[3,10], Hiroshi Arakawa[4], Takahiro Kakuta[2,3], Tada-aki Yamagishi[3], Jumpei Terakawa[5], Takiko Daikoku[5], Shin-ichi Horike [6], Sha Si[1,2], Kenta Kurayoshi[1], Chiaki Ito[1], Atsuko Kasahara[7], Yuko Tadokoro[1,2], Masahiko Kobayashi[1,2], Tsutomu Fukuwatari[8], Ikumi Tamai[4], Atsushi Hirao[1,2 ✉] & Tomoki Ogoshi [2,9 ✉]

Metabolic syndrome is associated with obesity, hypertension, and dyslipidemia, and increased cardiovascular risk. Therefore, quick and accurate measurements of specific metabolites are critical for diagnosis; however, detection methods are limited. Here we describe the synthesis of pillar[n]arenes to target 1-methylnicotinamide (1-MNA), which is one metabolite of vitamin B3 (nicotinamide) produced by the cancer-associated nicotinamide N-methyltransferase (NNMT). We found that water-soluble pillar[5]arene (P5A) forms host–guest complexes with both 1-MNA and nicotinamide, and water-soluble pillar[6]arene (P6A) selectively binds to 1-MNA at the micromolar level. P6A can be used as a "turn-off sensor" by photoinduced electron transfer (detection limit is $4.38 \times 10^{-6}$ M). In our cell-free reaction, P6A is used to quantitatively monitor the activity of NNMT. Moreover, studies using NNMT-deficient mice reveal that P6A exclusively binds to 1-MNA in crude urinary samples. Our findings demonstrate that P6A can be used as a biosensor to quantify 1-MNA in crude biological samples.

[1] Division of Molecular Genetics, Cancer and Stem Cell Research Program, Cancer Research Institute, Kanazawa University, Kakuma-machi, Kanazawa, Ishikawa 920-1192, Japan. [2] WPI Nano Life Science Institute (WPI-Nano LSI), Kanazawa University, Kakuma-machi, Kanazawa, Ishikawa 920-1192, Japan. [3] Graduate School of Natural Science and Technology, Kanazawa University, Kakuma-machi, Kanazawa, Ishikawa 920-1192, Japan. [4] Faculty of Pharmaceutical Sciences, Institute of Medical, Pharmaceutical and Health Sciences, Kanazawa University, Kakuma-machi, Kanazawa, Ishikawa 920-1192, Japan. [5] Institute for Experimental Animals, Advanced Science Research Center, Kanazawa University, Takara-machi, Kanazawa 920-8641, Japan. [6] Division of Functional Genomics, Advanced Science Research Center, Kanazawa University, Takara-machi, Kanazawa 920-8641, Japan. [7] Institute for Frontier Science Initiative, Kanazawa University, Kakuma-machi, Kanazawa, Ishikawa 920-1192, Japan. [8] Department of Nutrition, School of Human Cultures, The University of Shiga Prefecture, 2500 Hassaka, Hikone, Shiga 522-8533, Japan. [9] Department of Synthetic Chemistry and Biological Chemistry, Graduate School of Engineering Kyoto University, Kyoto 615-8510, Japan. [10] These authors contributed equally: Masaya Ueno, Takuya Tomita. ✉email: ahirao@staff.kanazawa-u.ac.jp; ogoshi@sbchem.kyoto-u.ac.jp

Metabolites are intermediate or end-chemical compounds of cellular metabolism[1]. In the metabolic process, organic compounds, including amino acids, sugars, nucleotides, and fatty acids, are broken down or modified to produce various small molecules, or they are used to construct an enormously diverse range of macromolecules such as proteins, carbohydrates, nucleic acids, and lipids that are necessary for cellular functions. Recent updates from comprehensive metabolite analyses identified over 110,000 metabolites in the human body (www.hmdb.ca)[2]. As a common clinical syndrome, metabolic syndrome (MS) poses a severe threat to human health[3]. MS includes symptoms of abdominal obesity, hypertension, and dyslipidemia. MS is also associated with risks of developing cardiovascular disease, type 2 diabetes, and various types of cancer; therefore, multiple metabolic abnormalities are observed within these diseases[4]. Thus, quick and accurate measurements of specific metabolites are critical to understand the pathological condition and evaluate the effects of therapeutic treatment.

For the detection of proteins that have relatively large molecular weights, various specific antibodies are available[5]. Through specific recognition by monoclonal antibodies, it is possible to quantify a particular peptide in crude biological samples or visualise its intracellular localisation using simple methods and equipment[5]. Detection of antigens by antibodies is not only specific but also very rapid. Some antibody-based detection methods require less than 30 min of experimental time; therefore, these methods are very advantageous in clinical diagnosis and large-scale studies[6].

In contrast, quantification methods for relatively low molecular weight metabolites are very limited. For the quantification of small compounds, high resolution $^1$H nuclear magnetic resonance (NMR) spectroscopy, high performance liquid chromatography (HPLC), and liquid chromatography-tandem mass spectrometry (LC-MS/MS) are commonly used in laboratories[7]. However, these methods require expensive equipment and skilled operators, and are not suitable for high-throughput analysis, such as drug screening and mass studies. Thus, development of sensors that can be used to specifically recognise a particular metabolite is essential for the development of rapid and accurate measurements of metabolites.

1-Methylnicotinamide (1-MNA) is a primary metabolite of nicotinamide, an amide form of vitamin B3 (niacin), and is produced by the enzymatic reaction of nicotinamide N-methyltransferase (NNMT (E.C. 2.1.1.1))[8]. NNMT catalyses the methylation of nicotinamide using the universal methyl donor S-adenosyl methionine (SAM), to produce S-adenosyl-L-homocysteine (SAH) and 1-MNA. The product of NNMT, 1-MNA, can be further oxidised by aldehyde oxidase into two related compounds, N1-methyl-2-pyridone-5-carboxamide (2py) and N1-methyl-4-pyridone-3-carboxamide (4py), and all three metabolites are eventually excreted in the urine[8]. Recently, it was reported that aggressive cancer cell lines have higher levels of 1-MNA because of increased NNMT activity[9]. This is supported by data showing NNMT is highly expressed in various cancers[8], and inhibition of NNMT results in suppression of cancer progression[10–16]. Therefore, it is important to develop NNMT inhibitors to eradicate cancer propagation in the clinic.

To detect 1-MNA in biological samples, we hypothesised that pillar[n]arenes can be used as "biosensors." Pillar[n]arenes, which were first reported by our group in 2008, are a family of pillar-shaped macrocyclic compounds that contain a para-bridge connection between the 1,4-dialkoxybenzene units[17]. Their pillar-shaped structures have an electron-rich cavity that has host–guest properties, and it is known that pillar[n]arenes form stable complexes with cations[18–23]. The diameter across the cavity of pillar[n]arenes is tuneable and is approximately 4.7 Å and 6.7 Å

for pillar[5]arene and pillar[6]arene, respectively[24]. Furthermore, the functional groups of pillar[n]arenes can be easily modified. Therefore, diverse pillar[n]arenes bearing the desired functional groups can be prepared conveniently[18–23]. Using the pillar[n] arenes, it is also possible to add a switchable function to compounds. For example, pillararene-based supramolecular polymers can be induced to create a cross-linked network in the presence of external stimuli such as heat, pH and $H_2S$[25]. Pillararene-based self-assembled amphiphiles can be induced to undergo conformational transformation between the gemini-type and bola-type by light irradiation[26].

Herein we report a water-soluble pillar[n]arene[27,28], pillar[6] arene carrying 12 carboxylate anions (P6A), that is used to specifically and quantitatively detect 1-MNA but not nicotinamide or other nicotinamide derivatives. $^1$H NMR, isothermal calorimetry (ITC), and fluorescence measurements indicateed that the detection limit of 1-MNA by P6A is $4.38 \times 10^{-6}$ M, which is 10 times smaller than that of pillar[5]arene with 10 carboxylate anions (P5A). P6A also has high selectivity as demonstrated by the fact that P6A do not form host–guest complexes with nicotinamide and 2py. Furthermore, P6A is also useful to detect 1-MNA in a crude biological sample. In addition, P6A is used to monitor NNMT activity in a cell-free enzymatic reaction, and our studies using a NNMT-deficient mouse model revealed that P6A exclusively binds and detects 1-MNA in crude urinary samples. Thus, our findings provide the basis of a novel strategy to establish a metabolite-specific biosensor using supramolecules.

## Results

**P6A selectively forms host–guest complexes with 1-MNA.** Nicotinamide is a pyridinecarboxamide, which is a pyridine where the hydrogen at the meta-position is replaced by a carboxamide group (Fig. 1a). 1-MNA is a pyridinium cation comprised of nicotinamide with a methyl group at the 1-position produced by the enzymatic reaction catalysed by NNMT. The cationic structure of 1-MNA is similar to dimethyl viologen (paraquat). Our group and others have shown that anionic P5A and P6A formed stable 1:1 host–guest complexes with paraquat with extremely high association constants of $8.2 \pm 1.7 \times 10^4$ $M^{-1}$ and $1.02 \pm 0.1 \times 10^8$ $M^{-1}$, respectively[27,28]. Thus, we hypothesised that the two water-soluble pillar[n]arenes P5A and P6A would be good candidates as biosensors for detection of 1-MNA (Fig. 1b–d).

The complexation of the pillar[n]arenes with 1-MNA was first studied by $^1$H NMR spectroscopy (Fig. 2). When P5A or P6A (1.5 mM) was added to 1-MNA (1.5 mM) in $D_2O$ solution, the proton peaks of 1-MNA broadened and shifted upfield in both cases (Fig. 2a, b), indicating host–guest complexation. Clear NOE cross peaks were observed between the aromatic proton signals of 1-MNA and aromatic proton signal of P6A (Supplementary Fig. 1), which also indicated the formation of a host-guest inclusion complex between P6A and 1-MNA. In the Job's plots of P6A and 1-MNA, the peak was observed at a molar fraction of $x_{guest} = 0.50$, indicating that the stoichiometry of the P6A–1-MNA complex was 1:1. The stoichiometry of the P5A-1-MNA complex was also 1:1, which was confirmed by the Job's plots (Supplementary Fig. 2).

Next, we investigated host–guest complexation of nicotinamide with P5A or P6A. Upon addition of P5A (1.5 mM) to nicotinamide (1.5 mM), broadening and shifting of the proton peaks of nicotinamide were observed (Fig. 2c), indicating host–guest complexation. The Job's plot revealed that the stoichiometry of the P5A–nicotinamide complex was 1:1 (Supplementary Fig. 3). However, no significant changes of the proton peaks from nicotinamide were detected when P6A was

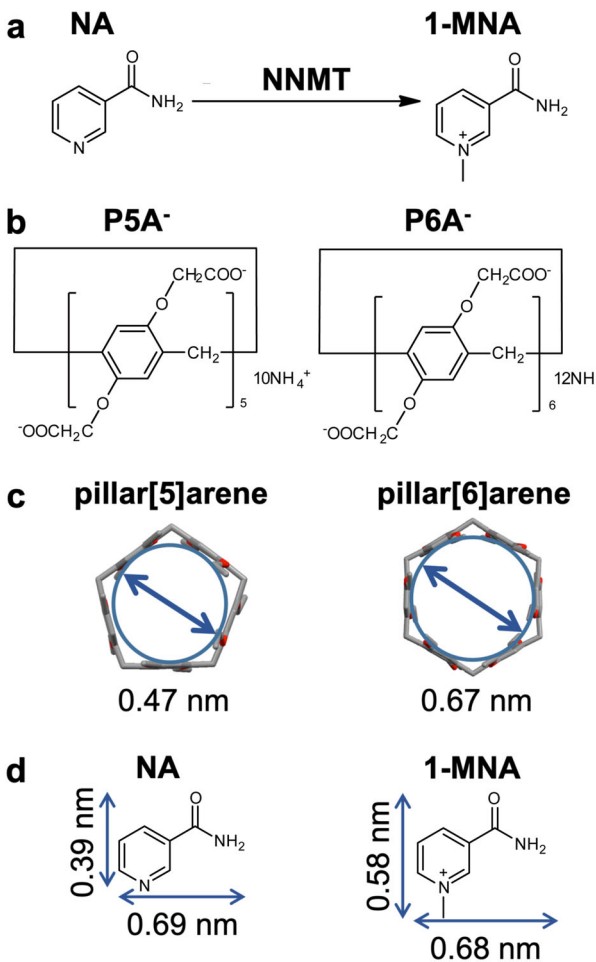

**Fig. 1 Chemical structures of molecules and complexes investigated in this work. a** NNMT catalyses the methylation of nicotinamide (NA) to produce 1-methylnicotinamide (1-MNA). **b** Structures of P5A and P6A. **c** Diameter of the inscribed circle within the cavity. **d** Molecular sizes of nicotinamide and 1-MNA.

mixed with nicotinamide (Fig. 2d), indicating that P6A did not form a host–guest complex with nicotinamide.

To determine the association constants ($K$) for the host–guest complexes, ITC measurements were performed (Fig. 3, Supplementary Figs. 2, 3). The $K$ values were determined as $1.28 \pm 0.19 \times 10^2$ $M^{-1}$ for the P5A–nicotinamide complex (Supplementary Fig. 3), $1.14 \pm 0.13 \times 10^3$ $M^{-1}$ for the P5A–1-MNA complex (Supplementary Fig. 2), and $8.05 \pm 0.96 \times 10^3$ $M^{-1}$ for the P6A–1-MNA complex (Fig. 3a). In the case of P6A and nicotinamide, only the heat of dilution was observed (Fig. 3b), indicating that the binding between P6A–nicotinamide was too small to determine the $K$ value. The ITC data were consistent with the NMR results described above. From these NMR and ITC results, it was concluded that P5A formed host–guest complexes with both nicotinamide and 1-MNA. There was no clear guest selectivity for P5A. In comparison, P6A had clear guest selectivity as it formed a host–guest complex with 1-MNA, and did not form a complex with nicotinamide. P6A was able to form a more stable host–guest complex with 1-MNA compared with P5A as demonstrated by the fact that the $K$ value for the P6A–1-MNA complex was eight times larger than that for the P5A–1-MNA complex (Fig. 3c). The critical specificity of molecular recognition is related to the pore size difference between P5A and P6A (Fig. 1c). Both P5A and P6A are comprised of electron donor dialkoxybenzene moieties that have carboxylate anions at both

rims. 1-MNA is the electron acceptor and has two pyridinium cations. Therefore, the donor–acceptor and ionic interactions both contribute to the host–guest complexation. The cavity size of P6A is more suited to the size of 1-MNA compared with P5A; thus, P6A formed a more stable host–guest complex with 1-MNA than P5A. P6A bound to 1-MNA and did not bind to nicotinamide, while P5A non-selectively formed relatively stable host–guest complexes with both 1-MNA and nicotinamide. The selectivity was also related to the cavity size differences of P5A and P6A. The estimated minor radius of nicotinamide is less than 60% of the pore size of P6A (Fig. 1d). Because of the small size of nicotinamide, it may be difficult for strong associations to form between P6A and nicotinamide. In comparison, the size of nicotinamide allowed it to fit more tightly into the cavity of P5A allowing P5A to capture it as a result of stronger associations.

To test the specificity of the molecular recognition of P6A, 2py, another nicotinamide metabolite, was used as a guest. Interestingly, limited shifts of the proton peaks of 2py were observed in the NMR spectrum (Supplementary Fig. 4), and it was not possible to calculate the $K$ value between P6A and 2py because the binding between P6A and 2py was too small to determine the $K$ value (Supplementary Fig. 5). 2py is a similar size to 1-MNA, but is not a cation; thus, P6A has a lower binding affinity to 2py compared with 1-MNA. These data clearly indicated that electrostatic interactions between the pyridinium cation of 1-MNA and carboxylate anions of P6A were also involved in the formation of a host–guest complex between P6A and 1-MNA.

**1-MNA quenches the fluorescence of P6A by PET**. To quantitatively assess the host–guest encapsulation between the pillar[n] arenes and the two metabolites, fluorescence titration experiments were performed (Fig. 4a, b, Supplementary Fig. 6). Upon addition of 1-MNA, the fluorescence intensity of both P5A and P6A dramatically decreased because of photoinduced electron transfer (PET)[27–32]. Even when an excess amount of nicotinamide (100 eq.) was added to P5A or P6A in solution, the quenching of emission from each pillar[n]arene was subtle. Nicotinamide is a neutral molecule, and thus no PET takes place upon addition of an excess amount of nicotinamide.

Detection limits were also determined from the fluorescence titration data (Fig. 4c, Supplementary Fig. 6c). There was a good linear relationship between the fluorescent intensity data at 325 nm and the micromolar concentrations of 1-MNA, indicating both pillar[n]arenes are useful to quantitatively detect relevant concentrations of 1-MNA. The linear equation for the fluorescent intensity–concentration relationship was found to be $y = 1901.5x + 1.0137$ ($R^2 = 0.9892$) for P5A, and $y = 3066.8x + 1.0105$ ($R^2 = 0.9762$) for P6A. The detection limits for 1-MNA were calculated to be $2.53 \times 10^{-5}$ M for P5A and $4.38 \times 10^{-6}$ M for P6A. Thus, the detection limit of 1-MNA by P6A was approximately six times smaller than that by P5A. These data demonstrated that P6A can be used as a "turn-off" fluorescent sensor for the sensitive and quantitative detection of 1-MNA in solution.

**P6A quantitatively monitors the enzymatic activity of NNMT**. To efficiently develop NNMT inhibitors, it is useful to establish a cell-free enzyme reaction system and a high-throughput measurement assay for detection of NNMT enzymatic activity. Since P6A was found to specifically bind to 1-MNA, we speculated that P6A could be used as a biosensor to monitor NNMT activity by the detection of 1-MNA in solution. Therefore, we developed a cell-free enzyme system to produce 1-MNA from nicotinamide. For the high-level expression and subsequent purification of NNMT protein in *Escherichia coli*, a glutathione-S-transferase (GST) fusion protein expression vector was used[33]. The GST-

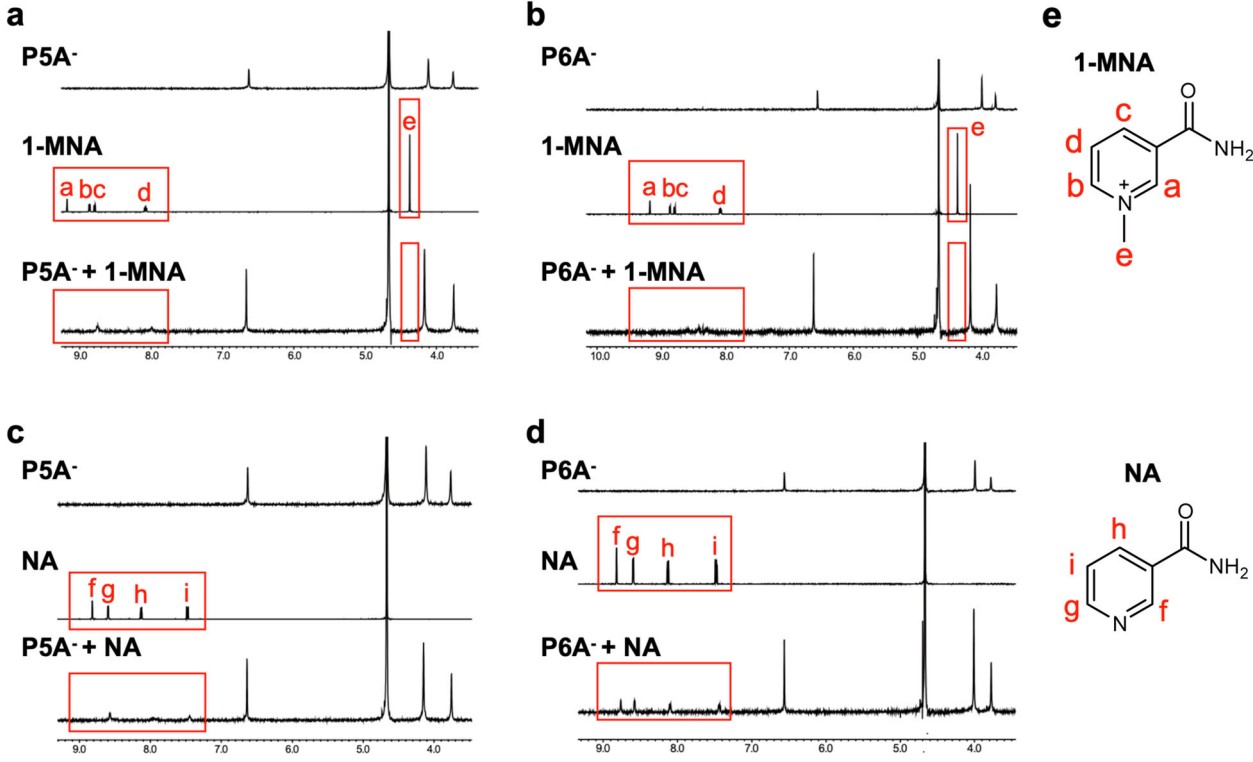

**Fig. 2 P5A forms host-guest complexes with both 1-MNA and nicotinamide, but P6A selectively binds to 1-MNA. a–d** Partial ¹H NMR spectra of the indicated complexes. The proton peaks of the guest molecule are indicated by the red rectangles: (**a**) P5A, 1-MNA, and P5A and 1-MNA, (**b**) P6A, 1-MNA, and P6A and 1-MNA, (**c**) P5A, nicotinamide (NA), and P5A and NA, and (**d**) P6A, NA, and P6A and NA. **e** Structures of the guest molecules.

fused *NNMT* (GST-NNMT) gene was transduced into *E. coli*, and then recombinant GST-fused protein was purified by a glutathione Sepharose column. We also generated an enzymatically inactive NNMT mutant that contained the Y20A mutation[34] (a tyrosine (Y) residue at position 20 in the NNMT protein was converted to an alanine (A)). To estimate the purity of each protein, we resolved the protein products using sodium dodecyl sulfate polyacrylamide gel electrophoresis (SDS-PAGE) and detected them by staining with Coomassie Blue. We observed a single major protein band in both the GST-NNMT and GST-NNMT Y20A samples (Supplementary Fig. 7), which suggested that the purities of the two proteins were sufficiently high for further analyses.

We next examined whether GST-NNMT could produce 1-MNA from nicotinamide in a cell-free reaction (Fig. 5). Purified proteins were incubated with nicotinamide and SAM at 37 °C, and 1-MNA was detected with LC-MS/MS. The increase of 1-MNA production was clearly observed in a substrate-dependent (nicotinamide and SAM) and time-dependent manner (Fig. 5a, b). The methylation of nicotinamide was strongly suppressed in the GST-NNMT Y20A mutant, and no 1-MNA production was observed from the GST construct alone. These data clearly indicated that 1-MNA production from nicotinamide is dependent on the enzymatic activity of the recombinant NNMT protein.

To test whether P6A could be used to detect 1-MNA produced by our cell-free system, a fluorescence measurement was performed. As shown in Fig. 5, the fluorescence intensity of P6A was quenched by 1-MNA in a dose-dependent manner. The percent inhibition (% inhibition; $(1 - (F/F_0)) \times 100$ was calculated. $F_0$ is the initial fluorescence intensity (before the enzymatic reaction, 0 min), $F$ is the fluorescence intensity for each experimental condition. We found a significant positive

correlation between the concentration of 1-MNA determined from the LC-MS/MS experiment and the values of the % inhibition obtained from the fluorescence measurements (Fig. 5c) ($R^2 = 0.9771$). Thus, the results obtained from the two methods agreed precisely.

We found that our biosensor P6A can detect 1-MNA produced by a cell-free NNMT enzyme reaction specifically and quantitatively. Therefore, we hypothesised that this system could also be used to monitor the suppression of methylation by NNMT inhibitors. Recently, JBSNF-000088 (6-methoxynicotinamide) was reported as an inhibitor for NNMT[35]. Addition of 6-methoxynicotinamide to the reaction inhibited it in a concentration-dependent manner (Fig. 5d). Furthermore, a similar result was obtained from fluorescence measurements using P6A (Fig. 5d). These data indicated that our biosensor P6A can be used to specifically and quantitatively monitor NNMT activity in the cell-free enzyme reaction system, and is applicable for the screening of NNMT inhibitors.

**P6A specifically and quantitatively detects urinary 1-MNA.** During the diagnosis of metabolic syndromes and the evaluation of therapeutic treatments, quantification of specific metabolites in tissues, blood, and/or urine is extremely important. Previous studies have reported that NNMT and/or its product, 1-MNA, are elevated not only in cancer but also in several metabolic syndromes[36–38]. Because of the specific and strong binding between P6A and 1-MNA, we expected that P6A could be used as a sensor for the detection of 1-MNA in crude biological samples. In animals, 1-MNA is excreted in the urine, and the concentration of urinary 1-MNA is highly dependent on the total uptake of nicotinamide from foods[39]. Recently, another group reported that the 1-MNA concentration in *Nnmt* knockout (KO) mice was

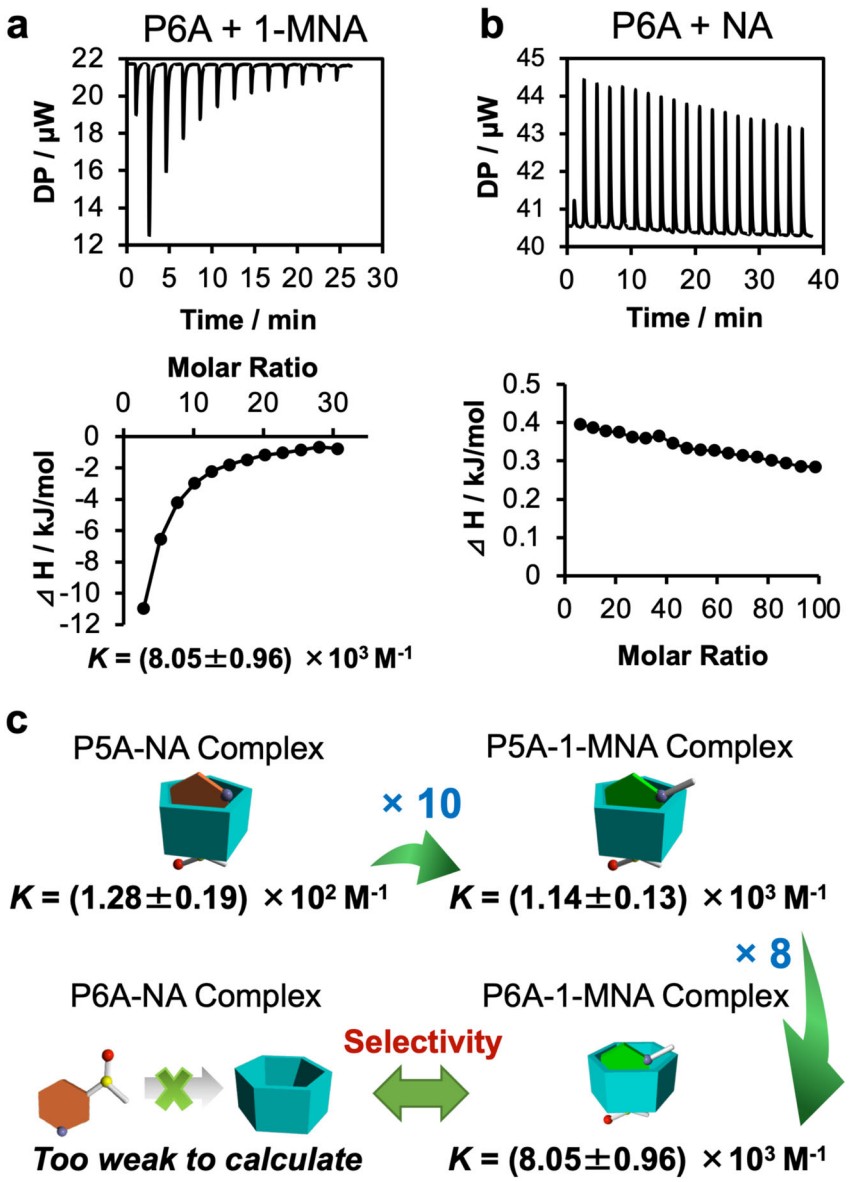

**Fig. 3 P6A selectively forms host–guest complexes with 1-MNA.** ITC experiments on the complexation (**a**) P6A with 1-MNA, and (**b**) P6A with nicotinamide (NA). **c** Summary of the association constants ($K$) for each host–guest complex.

reduced by >99% in plasma, liver, and visceral white adipose tissue[40], indicating that NNMT is the only enzyme responsible for the methylation of nicotinamide in animals. To test whether P6A can be used as a sensor for 1-MNA for crude biological samples, we generated a *Nnmt* KO mouse (Supplementary Fig. 8). In the *Nnmt* KO mouse, no unexpected alternative splicing variants of *Nnmt* mRNA nor functional Nnmt protein were detected in the liver (Supplementary Note 1, and Supplementary Figs. 9–13). *Nnmt* KO mice are viable, fertile and have a normal phenotype, implying that NNMT is not essential for murine development. Next, to confirm if 1-MNA production was deficient in our *Nnmt* KO mouse model, the serum concentration of 1-MNA was quantified by LC-MS/MS (Supplementary Fig. 8d). Consistent with the *Nnmt* deletion, 1-MNA in the serum of *Nnmt* KO mice was only minimally detected. Therefore, to test whether our biosensor could specifically detect 1-MNA in crude biological samples, urine from wild-type mice dosed with drinking water containing various doses of nicotinamide was collected. Furthermore, to evaluate the specificity of P6A, we collected urine from the *Nnmt* KO mice as a negative control as it does not

contain 1-MNA. Mass spectrometry analysis indicated that intake of nicotinamide led to an increase in the urinary excretion of 1-MNA in wild-type mice, while no 1-MNA excretion was detected in the *Nnmt* KO mice (Fig. 6a).

Urine has strong autofluorescence because it contains a number of natural fluorophores, most of which are tryptophan metabolites[41]. Thus, to decrease the background (autofluorescence signal), urine samples were subjected to a pre-column clean-up for removal of hydrophobic materials. As shown in Fig. 6, the fluorescence intensity of P6A was significantly quenched by wild-type derived urine samples (Fig. 6b, c). In addition, the fluorescence intensity was quenched by 1-MNA in a dose-dependent manner. The % inhibition was calculated where $F_0$ is the average of fluorescence intensity of P6A with urine derived from *Nnmt* KO mice ($n = 5$), and $F$ is the fluorescence intensity of P6A with each urinary sample. We found that a significant positive correlation existed between the concentration of 1-MNA determined by LC-MS/MS and the values of the % inhibition obtained from the fluorescence measurements (Fig. 6d) ($R^2 = 0.8576$). Furthermore, to validate the specificity of P6A to

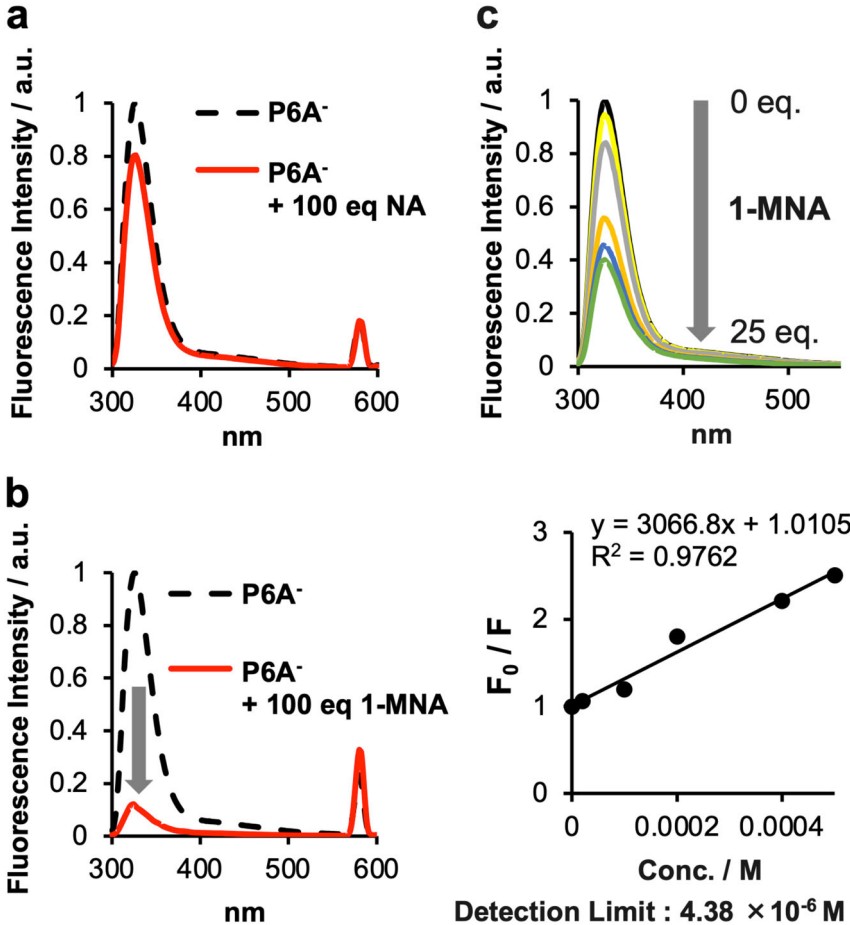

**Fig. 4 1-MNA quenches the fluorescence of P6A by photoinduced electron transfer. a, b** Fluorescence spectra of P6A with or without 100 eq. of (**a**) nicotinamide or (**b**) 1-MNA. **c** Fluorescence spectra of P6A with different concentrations of 1-MNA. Upon addition of 1-MNA, emission from P6A was quenched, indicating the formation of the P6A-1-MNA complex. Bottom: the linear least-squares analysis to calculate the detection limit of 1-MNA by P6A.

1-MNA, we diluted 1-MNA in *Nmnt* KO mouse-derived urine, and fluorescence measurements were performed (Supplementary Fig. 14, and Fig. 6d, dotted line). The correlation obtained for the concentration of 1-MNA determined by LC-MS/MS and the values of the % inhibition obtained from the fluorescence measurements of 1-MNA diluted in *Nmnt* KO urine were similar to the results above. These data clearly indicated that P6A can specifically and quantitatively detect 1-MNA in crude biological samples.

## Discussion

In conclusion, we found that P6A specifically forms host–guest complexes with 1-methylnicotinamide, and P6A can be used as a "turn-off" sensor by photoinduced electron transfer. P6A recognises the slight differences in structure present in nicotinamide, 1-MNA, and 2py. We also reported that P6A can be used to monitor the enzymatic activity of NNMT quickly and accurately. Therefore, our findings will contribute to establishing a high-throughput screen for NNMT inhibitors.

Although various fluorescent biosensors for cations (such as metal ions) have been known for many years, only a few compounds have been demonstrated to be sensors for human metabolites. While formation of a host–guest complex was observed using purified materials and solvents, our studies using NNMT-deficient mice revealed that P6A can specifically detect 1-MNA even in crude biological samples. Interestingly, it was reported that some supramolecular derivatives are applicable for biological

sensors. For example, Jang et al. reported that cucurbit[7]uril derivatives can selectively detect urinary amphetamine-type stimulants with a nanomolar detection limit[42]. It was also reported that guanidinium-modified calixarene can specifically detect lysophosphatidic acid, which is a biomarker for the early detection of ovarian and other gynaecological cancers[43]. Our findings provide a novel strategy for establishing metabolite-specific biosensors using supramolecules. Furthermore, it was recently demonstrated that a metallo-texaphyrin is a suitable magnetic resonance imaging contrast agent for detecting larger amyloid beta that is the main constituent of amyloid plaques in the brains of Alzheimer's disease patients[44]. Conjugation of an anticancer prodrug to gadolinium-texaphyrin enhanced its water solubility, and improved its anticancer activity[45].

In this research, we found that P6A can be used to specifically detect 1-MNA in urine. However, compared with the LC-MS/MS method (detection limit is <50 nM), the sensitivity of P6A to 1-MNA was very low, therefore, 1-MNA in blood plasma could not be detected using P6A (Supplementary Fig. 15). To improve the sensitivity of P6A for 1-MNA, further chemical modification is required. One significant and common problem in the detection of specific molecules by fluorescence biosensors is "autofluorescence", that is, fluorescence from inherent components of biological samples. To increase the fluorescent signal of the P6A, surface modification by pillararene may be useful to develop highly-sensitive sensor materials. Previously, we reported that immobilised multilayer pillar[5]arene films could efficiently absorb a guest molecule, para-dinitrobenzene (*p*-DNB), and the

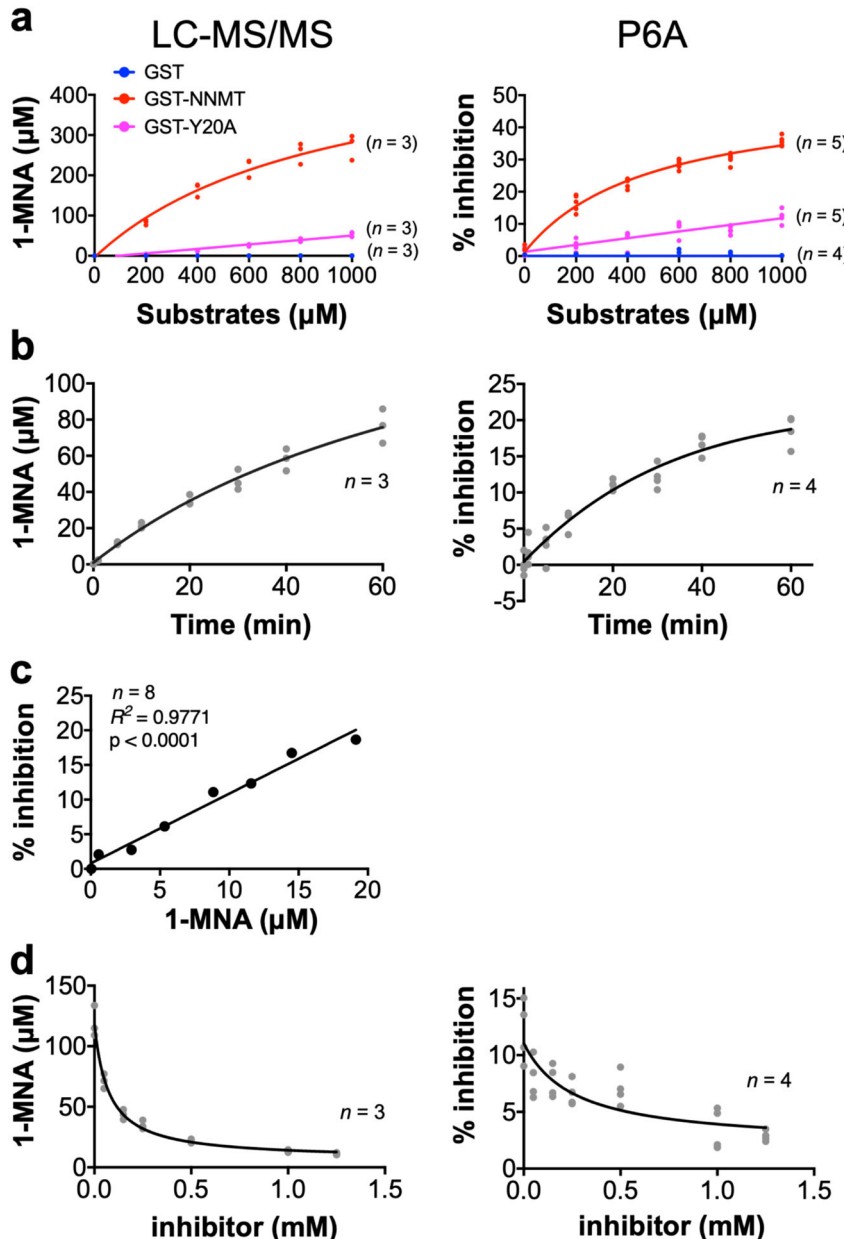

**Fig. 5 P6A specifically and quantitatively monitors the enzymatic activity of NNMT in a cell-free reaction system. a, b** 1-MNA is produced by recombinant NNMT from nicotinamide in a (**a**) substrate-dependent and (**b**) time-dependent manner. **c** The linear correlation between the concentration of 1-MNA obtained from LC-MS/MS and the % inhibition obtained from the fluorescent measurements. Line represents the linear regression calculation ($n = 8$). **d** Inhibition of 1-MNA production by a NNMT inhibitor. **a, b, d** The left panels show the quantification data using LC-MS/MS. The right panels are the data from fluorescence spectra with P6A.

adsorbed amount of *p*-DNB exponentially increases with an increasing number of pillar[5]arene-layers[46]. Further experiments will help to improve the sensitivity and specificity of the biosensors, and ultimately, this work should contribute to the development of low-cost, easy, and rapid methods for detection of human metabolites for diagnosis.

## Methods

**Chemicals**. All chemical reagents were purchased commercially, and used without further purification.

**Synthesis of pillar[5]arene and pillar[6]arene**. Pillar[5]arene (P5A) and pillar[6]arene (P6A) were synthesised according to previous papers[27,28].

**NMR measurements**. Solution [1]H NMR spectra were recorded at 500 MHz with a JEOL-ECA500 spectrometer (JEOL, Tokyo, Japan).

**Isothermal calorimetry measurements**. Isothermal calorimetry (ICT) experiments were performed on a MicroCal PEAQ-ITC (Malvern Panalytical, Malvern, UK). The P5A or P6A solution at 200 μM was placed in the sample cell, and a 6–60 mM substrate solution was loaded in the injection syringe.

**Fluorescence measurements**. Fluorescence spectra were recorded on a Hitachi F-7000 fluorescence spectrometer (Hitachi High-Tech Science Corp., Tokyo, Japan) at room temperature. For the detection of 1-MNA produced in a cell-free system, 150 μL of P6A (final concentration is 20 μM in water) was added to 50 μL of sample (see below), and placed into wells of a 96-well plate, and then the fluorescence spectrum was immediately measured using a TECAN Spark (TECAN, Männedorf, Switzerland). For the quantification of urinary 1-MNA, 5 μL of pre-cleaned sample (see below) was added to 195 μL of P6A (final concentration was 200 μM).

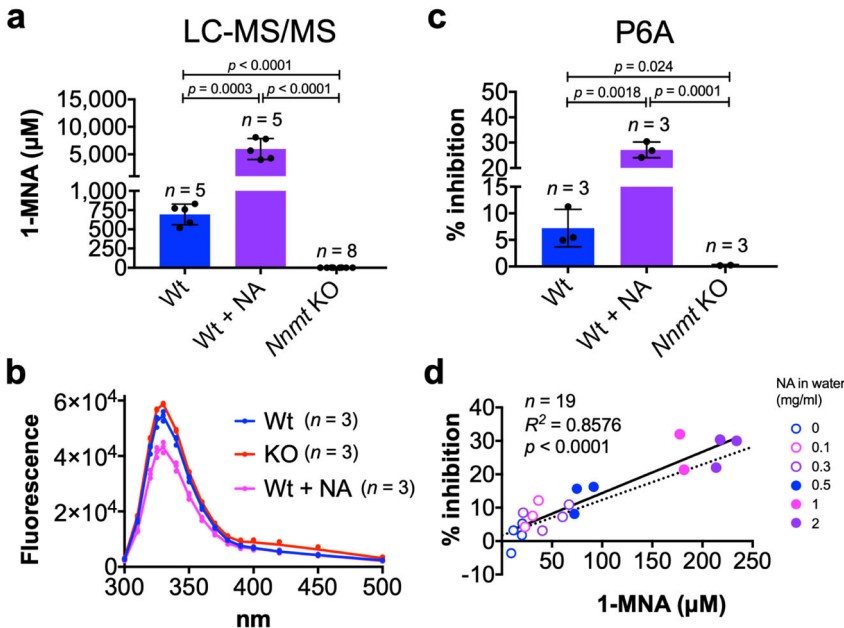

**Fig. 6 P6A specifically and quantitatively detects urinary 1-MNA. a** Concentration of urinary 1-MNA quantified by LC-MS/MS. **b** Fluorescence spectra of P6A for detection of 1-MNA from urinary samples. **c** Fluorescence of P6A at 325 nm for detection of 1-MNA from urinary samples. **a–c** In the nicotinamide treatment group (Wt + NA), the drinking water contained 2 mg/mL of nicotinamide. **d** The linear correlation between the concentration of 1-MNA obtained from LC-MS/MS and the % inhibition obtained from the fluorescence measurements. Line represents the linear regression calculation ($n = 19$). Wild-type mice were treated with drinking water containing various doses of nicotinamide (from 0 to 2 mg/mL). The dotted line is derived from Supplementary Fig. 14. Error bars represent mean ± s.d. Statistical significance was determined using an unpaired two-tailed Student's t-test.

**Construction of the NNMT expression vector**. For the generation of recombinant NNMT protein, a glutathione-*S*-transferase (GST) fusion protein expression system was used. To construct an expression vector for the production of the GST-NNMT fusion protein in *E. coli*, a DNA segment that codes NNMT protein was amplified. First, to synthesise the complementary DNA (cDNA), the reverse transcription reaction was performed using the QuantiTect Reverse Transcription Kit (QIAGEN, Hilden, Germany). The total RNA was isolated with the RNeasy Mini Kit according to the manufacturer's instructions (QIAGEN) from the human leukaemia cell line K562 and cDNA was synthesised from 1 µg of total RNA. To amplify the *NNMT* gene, polymerase chain reaction (PCR) was performed using DNA Taq polymerase (Ex Taq; Takara, Kusatsu-shi, Japan). Synthesised cDNA was used as the template DNA in the PCR as described below. The pair of PCR primers was purchased from Thermo Fisher Scientific (Waltham, MA). The sense PCR primer was derived from the 5′ nucleotide sequence: [5′-GAA TTC CAT GGA ATC AGG CTT CAC CTC CAA-3′], and the antisense primer was based on the 3′ end of the human *NNMT* gene: [5′-CTC GAG TCA CAG GGG TCT GCT CAG CTT CCT-3′]. For cloning, each primer contained an *Eco*RI or *Xho*I restriction enzyme recognition sequence on the ends of the primers (underlined in the sequences above). The PCR was subjected to initial denaturation at 94 °C for 2 min followed by 40 cycles of 94 °C (30 s), 55 °C (30 s), 72 °C (1 min) and a final extension at 72 °C (5 min). The DNA fragment (NNMT coding region) amplified by PCR and the pGEX-4T-3 plasmid (GE Healthcare, Chicago, IL) were digested with *Eco*RI and *Xho*I. After digestion, both DNAs were linked together using T4 DNA ligase for construction of pGEX-NNMT.

**Mutagenesis**. To introduce the mutation in the *NNMT* gene in pGEX-NNMT (see above), PCR based site-directed mutagenesis was performed using the PrimeSTAR Mutagenesis Basal Kit (Takara). The primer used for mutagenesis had the sequence 5′-TAA GCC ATT TTA ACC TCG GGA ATG CCC TAG AAA AAT ATT ACA AGT TTG GT-3′. This sequence corresponds to nucleotides 35 through 84 of the human NNMT coding sequence except for nucleotides 59 and 60 where T and A replaced G and C, respectively (underlined in the sequence above). These base changes converted the codon for tyrosine at position 20 in the NNMT protein to the codon for alanine (Y20A). Plasmids carrying the Ala-20 mutation (pGEX-NNMT Y20A) were identified by direct DNA sequencing of the mutated region of the DNA.

**Purification of NNMT recombinant protein**. The constructs (pGEX-4T-3, pGEX-NNMT or pGEX-NNMT Y20A) were transformed into *E. coli* DH5α (Toyobo, Osaka, Japan). Transformants were grown in LB medium containing ampicillin (0.1 mg/mL), at 37 °C in a shaking incubator. Expression of the fusion protein was induced by the addition of 0.1 mM isopropyl β-D-thiogalactopyranoside (IPTG)

(Fujifilm Wako Chemical Corp., Osaka, Japan). GST and GST fusion proteins (GST-NNMT and GST-NNMT Y20A) were purified from bacterial lysate by a glutathione Sepharose column (GE Healthcare). Proteins were eluted from the glutathione column by addition of excess reduced glutathione. To remove the glutathione from the buffer, the eluted protein was dialysed against PBS (phosphate-buffered saline).

**Gel electrophoresis**. Proteins were dissolved in SDS sample buffer containing 60 mM Tris–HCl, 2% SDS, 10% glycerol, and 5% 2-mercaptoethanol. The samples were heated at 95 °C for 5 min and separated by electrophoresis on a polyacrylamide gradient (4%–20%) gel (Fujifilm Wako Chemical Corp.). Proteins were visualised using Coomassie Brilliant Blue R-250 staining.

**Cell-free nicotinamide methylation assay**. For the NNMT methyltransferase reactions, the buffer contained 20 mM Tris (pH 8.0), 50 mM NaCl, 1 mM EDTA, 3 mM MgCl$_2$, and 1 mM DTT. Buffer also contained enzyme, nicotinamide (from 0 to 1 mM), and *S*-adenosyl-L-methionine (SAM; from 0 to 1 mM) at the concentrations indicated in each figure. Reactions were carried out at 37 °C for 3 h or the indicated time in a LoBind tube (Eppendorf, Hamburg, Germany). After the enzymatic reaction, 20 volumes of ethanol were added to the sample tube to precipitate the protein, which was then removed by centrifugation at 12,000 rpm for 10 min. The ethanol mixture was transferred to a new tube and evaporated. Once all the ethanol was evaporated, the pellet was resuspended in the same original volume of ultrapure water (Merck Millipore, Burlington, MA).

**Experimental animals**. All animal care was performed in accordance with the guidelines established by Kanazawa University for animal use. All animal experiments were approved by the Animal Care and Use Committee of Kanazawa University (AP-204142). C57BL/6 mice female were purchased from Japan SLC (Hamamatsu, Japan). Mice between 12 and 20 weeks old were provided drinking water containing 0 to 2 mg/mL of nicotinamide ad libitum. After 2 days of drinking nicotinamide-containing water, urine was collected individually. For collection of the urine, each mouse was held over a Petri dish and urination was stimulated by application of gentle trans-abdominal pressure over the bladder to overcome normal urethral pressure[47].

**Establishment of the *Nnmt* KO mice**. CRISPR-mediated removal of exon 2 of the *Nnmt* gene from the genomic DNA was performed in fertilised eggs. In the public VEGA database (http://vega.archive.ensembl.org/index.html), five alternative splicing variants of *Nnmt* mRNA were found. Based on the Human And Vertebrate Analysis aNd Annotation (HAVANA), two splicing variants of *Nnmt* mRNA

(Nnmt-001 and Nnmt-002) have protein coding sequences. While the *Nnmt-001* codes the full protein coding sequence of Nnmt protein and *Nnmt-002* lacks a large part of exon 3, both mRNAs contain the whole sequence of exon 2. The exon 2 deletion caused a frameshift in the open reading frame and knockout (KO) gene. For preparation of the gRNAs targeting the mouse *Nnmt*, we designed gRNA#1 (5′-GGA TTG TGC CAT TTT TGT TG-3′) and gRNA#2 (5′-TGT ACC ACC AGA GTA CTA TT-3′) to target the 3′ end of intron 1 and the 5′ end of intron 2 of *Nnmt*, respectively. Each CRISPR RNA (crRNA) to generate the gRNAs was synthesised by Integrated DNA Technologies (IDT; Coralville, IA) as an Alt-R® CRISPR crRNA product.

The fertilised pronuclear-stage embryos were prepared by in vitro fertilisation in human tubal fluid medium (HTF; ARK Resource; Kumamoto, Japan) with sperm from male C57BL/6J mice (Japan SLC, Inc.) and oocytes from superovulated female C57BL/6J mice[48]. The complexes of crRNAs (2 μM each), tracrRNA (IDT, 4 μM) and Cas9 protein (IDT, 100 ng/μL) suspended in Opti-MEM (Thermo Fisher Scientific) were then introduced to the pronuclear-stage embryos by electroporation[49]. After electroporation, the embryos were washed and cultured in potassium simplex optimisation medium (KSOM; ARK Resource) overnight. The two-cell embryos were transferred to pseudo-pregnant recipient ICR mice (CLEA Japan, Inc., Tokyo, Japan)[49]. Pups from transplanted embryos were considered as the F0 generation. The genotypes of the F0 generation were analysed by PCR with genomic DNA extracted from tail biopsies of the animals. To amplify the DNA fragments containing the genome edited region, the genotyping PCR was performed with the primers 5′-TTT ACC CAC TGA CCG TCT CC-3′ and 5′-GAC CAT CAC TCC AGG CCT TA-3′, and GoTaq (Promega, Madison, WI). The PCR cycle was subjected to initial denaturation at 95 °C for 2 min followed by 35 cycles of 95 °C (30 s), 58 °C (30 s), 72 °C (1 min) and a final extension at 72 °C (7 min). The PCR products were electrophoresed on a 2% agarose gel and DNA was visualised by ethidium bromide staining.

To validate the CRISPR editing by Sanger sequencing, the PCR products were gel-purified using a gel extraction kit (QIAGEN) and cloned into the pGEM-T Easy vector (Promega) by TA cloning. Plasmid DNAs were submitted for Sanger sequencing to determine the DNA sequences in the edited mice. Among the two founder (F0) mice obtained, three alleles carried the deletion that removed the region flanked by the sites targeted by the two gRNAs, gRNA#1 and gRNA#2. Then, one mutant mouse (#2) that carried the *Nnmt* heterozygous deletion allele was backcrossed with the wild-type mice to obtain the F1 generation to purify the *Nnmt* mutation allele. The backcross also enabled us to eliminate phenotypic variation, which might have arisen because of allelic heterogeneity or mosaicism in the F0 generation. After three generations of backcrosses and inbreeding, we generated *Nnmt* homozygous-deficient (KO) mice by crossing *Nnmt* heterozygous-deficient (Ht) mice.

**Pre-treatment of biological samples for detection of 1-MNA**. For removal of hydrophobic molecules from the urine and plasma samples, we used MonoSpin C18 columns (GL Science, Tokyo, Japan), which are centrifugal monolithic silica mini spin-columns. The spin column was conditioned with 0.2 mL of methanol at 5000 rpm for 1 min followed by 0.2 mL of PBS at 5000 rpm for 1 min. A total of 50 μL of the urine sample was then applied to the conditioned column and centrifuged for 1 min at 5000 rpm. This step was repeated two more times. Then the flow-through was further purified by ethanol precipitation (see above).

**Western blot**. Liver tissues were lysed in a lysis buffer (50 mM Tris-HCl (pH 8.0), 150 mM NaCl, 1% Nonidet P-40, 0.5% sodium deoxycholate, 0.1% SDS, and 1× Complete Mini Protease Inhibitor Cocktail (Sigma-Aldrich)). The samples were separated by electrophoresis on a polyacrylamide gradient gel (see above), and transferred to 0.45 mm PVDF membranes (Millipore, Billerica, MA, USA). Membranes were blocked in 5% skim milk in TBST (137 mM NaCl, 2.68 mM KCl, 25 mM Tris (pH7.4), and 0.05% Tween 20), incubated with a primary antibody in immuno-enhancer (Fujifilm Wako Chemical) overnight at 4 °C, and then incubated with horseradish peroxidase-conjugated anti-mouse immunoglobulin G antibody (GE Healthcare) before detection with ECL Prime (GE Healthcare, Piscataway, NJ, USA). Primary antibodies recognising the following proteins were used: Nnmt (OTI3D8) (abcam, # ab119758), and β-actin (Sigma-Aldrich, #A5441).

**Reverse transcription (RT)-PCR**. Total RNA was extracted from mouse liver tissues using the RNeasy Mini Kit (Qiagen), and total RNA was reverse transcribed into cDNA using a QuantiTect Reverse Transcription Kit (QIAGEN). To amplify the mouse *Nnmt* cDNA, PCR was performed using a Tks Gflex DNA polymerase (Takara). Synthesised cDNA was used as the template DNA for this PCR. The PCR was subjected to initial denaturation at 94 °C for 2 min followed by 30 cycles of 94 °C (30 s), 60 °C (30 s), 68 °C (30 s), and a final extension at 68 °C (5 min). The sequences of primers used were as follows: *Nnmt* Fw1, 5′-ACC TCC AAG GAC ACT TAT CTA AGT C-3′; *Nnmt* Rev1, 5′-TCA TGT AGT AGC TAC TCT TCA GGG C-3′; *Nnmt* Fw2, 5′-GAG CAT CAG GCT GGT GCA CGG AGC T-3′; *Nnmt* Rev2, 5′-AGC AGG CCT TTA ATA GGG ATC ACA G-3′; *Gapdh* Fw, 5′-TTC CAG TAT GAC TCC ACT CAC G-3′; *Gapdh* Rev, 5′-AGA CTC CAC GAC ATA CTC AGC A-3′. After the PCR reaction, Ex Taq (Takara) was added to the PCR reaction mix, and it was incubated at 72 °C for 25 min for addition of a 3′ A overhang directly to the 3′-ends of a blunt-ended DNA fragment. PCR products were purified by the QIAquick Gel Extraction Kit (Qiagen) from the agarose gel, and subcloned by using the TOPO TA Cloning Kit (Thermo Fisher Scientific). For the direct sequencing, PCR products were purified using a QIAquick-spin PCR purification kit (Qiagen).

**Quantitation of 1-MNA by LC-MS/MS**. The concentration of 1-MNA was determined using an LC-MS-8050 triple quadrupole LC-MS/MS (Shimadzu, Kyoto, Japan) coupled to an LC-30A system (Shimadzu). Chromatography was performed on an InertSustain amide column (ID 2.0 mm × 50 mm; GL Sciences) at 40 °C by means of step-gradient elution (flow rate, 0.4 mL/min) as follows: 0 to 2.0 min, 13% A/87% B; 2.0 to 2.5 min, 13% A/87% B to 50% A/50% B; 2.5 to 4.0 min, 50% A/50% B; 4.0 to 4.5 min, 50% A/50% B to 13% A/87% B; and 4.5 to 7.0 min, 13% A/87% B. A was water containing 0.1% formic acid and B was acetonitrile containing 0.1% formic acid. The mass numbers of the molecular and product ions for each compound were as follows: 1-MNA (137.2 → 94.2, CE − 22.0 V) and 1-MNA-d3 (Toronto Research Chemicals, North York, Canada) (140.0 → 97.2, CE −22.0 V). Lab solutions software (version 5.89, Shimadzu) was used for data manipulation. The detection limit was 10 ng/mL for each compound.

**Statistical analysis**. Statistical calculations were performed in GraphPad Prism. Values were compared using a two-tailed, unpaired Student's t-test. Statistical tests, n values and *p* values are located in the figures and/or legends. A *p* value of <0.05 was considered statistically significance. No statistical methods were used to pre-determine sample size.

**Reporting summary**. Further information on research design is available in the Nature Research Reporting Summary linked to this article.

## Data availability

All data supporting the findings of this study are available within the article and its Supplementary Information files or are available from the corresponding authors upon reasonable request. All biological materials are also available from the corresponding authors upon reasonable request. Uncropped pictures of gels and blots are shown in the Supplementary Fig. 16.

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

## Acknowledgements

M.U. was supported by a Grant-in-Aid for Scientific Research (C) (17K09919). A.H. was supported by a Grant-in-Aid for Scientific Research (A) (19H01033) from the Ministry of Education, Culture, Sports, Science, and Technology (MEXT), Japan and a Grant-in-Aid for Project for Cancer Research and Therapeutic Evolution (P-CREATE) (19cm0106104h0004) from the Japan Agency for Medical Research and Development (AMED). T.O. was supported by JST CREST (JPMJCR18R3) and a Grant-in-Aid for Scientific Research (A) (19H00909). This work was supported by a WPI-NanoLSI Transdisciplinary Research Grant from Kanazawa University. We thank Dr. Tsuyoshi Taniguchi (Kanazawa University) for 2D NOESY study. NanoLSI is supported by the World Premier International Research Center Initiative (WPI), MEXT, Japan. We thank Renee Mosi, PhD, from Edanz Group (https://en-author-services.edanzgroup.com/) for editing a draft of this manuscript.

## Author contributions

M.U. and T.T. conceived the hypothesis, designed and performed the experiments and analysed the data. H.A., T.K., T.Y., S.S., K.K., C.I., A.K., Y.T., M.K., T.F., and I.T. carried out the experiments. J.T., T.D., and S.I.H. generated the Nnmt KO mouse. A.H. and T.O. conceived the hypothesis and directed the project. M.U., A.H., and T.O. prepared the manuscript, which all authors edited and approved.

## Competing interests

The authors declare no competing interests.
