## [Peer Review File · Communications Chemistry]

Reviewers' comments:

Reviewer #1 (Remarks to the Author):

In this very nice manuscript, the authors synthesize pillar[n]arenes to target 1-methylnicotinamide (1-MNA), which is one metabolite of vitamin B3 (nicotinamide) produced by the cancer-associated nicotinamide N-methyltransferase (NNMT). According to their results, water-soluble pillar[5]arene (P5A) forms host-guest complexes with both 1-MNA and nicotinamide, and water-soluble pillar[6]arene (P6A) selectively binds to 1-MNA at the micromolar level. P6A can be used as a "turn-off sensor" by photoinduced electron transfer. In the cell-free reaction, P6A is used to quantitatively monitor the activity of NNMT. Moreover, studies using NNMT-deficient mice reveal that P6A exclusively binds to 1-MNA in crude urinary samples. These findings demonstrate that P6A can be used as a nanosensor to quantify 1-MNA in crude biological samples. Overall, the results presented in this manuscript are of great interest to the readers of Communications Chemistry. Furthermore, this manuscript has enough scientific value and novelty. Therefore, I suggest the acceptance for publication after the following minor points are addressed:

1. Some of the following recently published papers about pillararenes should be cited: Chem. Commun. 2018, 54, 12230–12233; Nanoscale 2017, 9, 8913–8917.
2. In order to further investigate the host-guest complexation between the water-soluble pillararenes and guest molecules here, 2D-NOESY NMR experiment should be provided.
3. Line 102–104 of page 6, the sentence "Furthermore, the functional groups of pillar[n]arenes can be easily modified; therefore, diverse pillar[n]arenes bearing the desired functional groups can be prepared conveniently" should be changed to "Furthermore, the functional groups of pillar[n]arenes can be easily modified. Therefore, diverse pillar[n]arenes bearing the desired functional groups can be prepared conveniently."
4. Line 107 of page 7, the full name of "NMR" should be provided for the first time use in the main text.
5. Line 108 of page 7, the word "indicate" should be "indicated".
6. Line 114 of page 7, the word "reveal" should be "revealed".

Reviewer #2 (Remarks to the Author):

In this paper, the authors first synthesized pillar[5]arenes and pillar[6]arenes and used them to quantify 1-MNA in vitro. The authors then generated NNMT deficient mice using CRISPR/Cas9 and demonstrated that pillar[6]arenes can be used to measure 1-MNA concentration in urine.

Upon the editor's request, the reviewer mainly evaluates the animal experiment part.

1. The animal experiments are appropriately performed. The reviewer recommends authors to disclose the approval number from the Institutional Animal Care and Use Committee (line 415-6).
2. The authors measured 1-MNA concentration in serum and showed it is about 1,000 times less than in urine (Fig s7). The reviewer would like to know if the P6A can be used in such a low concentration range.
3. Since Nnmt1 KO samples are absolute negative control, the authors should dilute 1-MNA in KO mouse urine and use for P6A validation.

Minor

4. The authors removed exon 2 that cause frameshift mutation. The reviewer agrees that the loss of exon2 disrupts gene function, but unexpected alternative splicing variants sometimes appear. The authors should show there are no alternative splicing variants nor functional NNMT proteins in KO mice.
5. While line 271 says "various doses of nicotinamide", there is only one column presented without any information in figure 6.

Reviewer #3 (Remarks to the Author):

I quite like the idea of this manuscript, which is to use pillar[n]arenes to target 1-methylnicotamide. Despite that enthusiasm, there are a number of relatively minor issues that reduce my overall enthusiasm for the manuscript and will need to be addressed before I can recommend publication. These include:

1. The abstract begins with information about metabolic syndrome and the associated health risks. While this is interesting information, it is only tangentially related to the main point of the manuscript, which focuses on supramolecular complexation inside pillar[n]arene guests. As such, this information should be eliminated from the abstract and moved to the introduction section exclusively.
2. Moreover, when the abstract indicates that P6A can be used as a "turn-off" sensor, the authors should provide quantitative information about how effective this sensor is.
3. The abstract also uses the term "nanosensor" without defining what this means or why P6A would qualify.
4. In the introduction, the information about antigen-antibody binding needs to have a supporting reference.
5. Since pillar[n]arenes have been reported since 2008, it is not clear that the adjective "new" should be used for their description, as the authors do in the introduction.
6. Also in the introduction, the authors claim that the cavity size of pillar[5]arene and pillar[6]arene are 4.7 and 6.7 angstroms. I assume the authors are referring to diameter across the cavity; if so, they should indicate that more explicitly.
7. In the introduction, when the authors refer to the use of "NMR," they should be more explicit and refer to "NMR spectroscopy" and in particular differentiate between proton and carbon NMR.
8. The authors claim that paraquat is "similar in structure" to 1-MNA. They should clarify by what metrics they are assessing this similarity.
9. I am concerned about the lack of control experiments reported herein. The authors claim that the reason for selectivity of the pillar[n]arene is related to the size of the cavity, but do not exclude that ionic interactions may be solely responsible for the observed effects. More information should be provided to explain how the authors are accounting for favorable host-guest electrostatic interactions.
10. The authors claim that photoinduced electron transfer is occurring between the guest and the host. It is far from clear that photoinduced electron transfer is in fact the mechanism by which such fluorescence decreases are observed. The authors should justify the proposed mechanism and/or provide relevant supporting references.
11. In the section focused on detection limits, the authors should provide information about other systems that detect these metabolites and what the limits of detection for these systems are.
12. The fact that the system works in unpurified urine is certainly interesting, but urine is actually a fairly simple biological fluid. Have the authors conducted any testing in more complex biological systems? If so, they should include results here. If not, they should consider doing so and/or at a minimum provide a hypothesis as to how well the system would work.

October 14th, 2020

Dear Reviewers,

I'm sending the revised manuscript of our paper entitled:

“Pillar[6]arene is a biosensor for quantitative detection of a vitamin metabolite in crude biological samples” (According to the reviewer's comment, we changed the title).

According to three reviewer's comments, we have revised our manuscript. We would like to explain these corrections point-by-point toward each reviewer as follows:

Reviewer #1 (Remarks to the Author):

In this very nice manuscript, the authors synthesize pillar[n]arenes to target 1-methylnicotinamide (1-MNA), which is one metabolite of vitamin B3 (nicotinamide) produced by the cancer-associated nicotinamide N-methyltransferase (NNMT). According to their results, water-soluble pillar[5]arene (P5A) forms host-guest complexes with both 1-MNA and nicotinamide, and water-soluble pillar[6]arene (P6A) selectively binds to 1-MNA at the micromolar level. P6A can be used as a “turn-off sensor” by photoinduced electron transfer. In the cell-free reaction, P6A is used to quantitatively monitor the activity of NNMT. Moreover, studies using NNMT-deficient mice reveal that P6A exclusively binds to 1-MNA in crude urinary samples. These findings demonstrate that P6A can be used as a nanosensor to quantify 1-MNA in crude biological samples. Overall, the results presented in this manuscript are of great interest to the readers of Communications Chemistry. Furthermore, this manuscript has enough scientific value and novelty. Therefore, I suggest the acceptance for publication after the following minor points are addressed:

Reply: We are very grateful for the reviewer 1's comments and valuable suggestions.

1. Some of the following recently published papers about pillararenes should be cited: Chem. Commun. 2018, 54, 12230–12233; Nanoscale 2017, 9, 8913–8917.

Reply: The recommended references are very valuable to our work. We have referenced them in the introduction section.

Revision in the “introduction” section: We have inserted the sentences below.

Using the pillar[n]arenes, it is also possible to add a switchable function to compounds. For example, pillararene-based supramolecular polymers can be induced to create a cross-linked network in the presence of external stimuli such as heat, pH, and H₂S²⁵. Pillararene-based self-assembled amphiphiles can be induced to undergo conformational transformation between the gemini-type and bola-type by light irradiation²⁶.

2. In order to further investigate the host-guest complexation between the water-soluble pillararenes and guest molecules here, 2D-NOESY NMR experiment should be provided.

Reply: We are very grateful to this valuable suggestion. In the revised manuscript, we have provided the data for the 2D-NOESY in the “Supplementary Information”. Clear NOE cross peaks were observed between the aromatic proton signals of 1-MNA and the aromatic proton signal of P6A, which indicated the formation of a host-guest inclusion complex between P6A and 1-MNA. We have also modified the text in the “Results” (see below).

Revision in the “Results”:

Clear NOE cross peaks were observed between the aromatic proton signals of 1-MNA and aromatic proton signal of P6A (Supplementary Figure 1), which also indicated the formation of a host-guest inclusion complex between P6A and 1-MNA.

3. Line 102–104 of page 6, the sentence “Furthermore, the functional groups of pillar[n]arenes can be easily modified; therefore, diverse pillar[n]arenes bearing the desired functional groups can be prepared conveniently” should be changed to “Furthermore, the functional groups of pillar[n]arenes can be easily modified. Therefore, diverse pillar[n]arenes bearing the desired functional groups can be prepared conveniently.”

Reply: Thank you for this correction. We have changed these sentences.

4. Line 107 of page 7, the full name of “NMR” should be provided for the first time use in the main text.

Reply: Thanks to the referee for this correction. We have added the words, “¹H nuclear magnetic resonance”.

5. Line 108 of page 7, the word “indicate” should be “indicated”.

Reply: Thanks to the referee for this correction. We have changed this spelling mistake.

6. Line 114 of page 7, the word “reveal” should be “revealed”.

Reply: Thanks to the referee for this correction. We have changed this grammatical error.

Reviewer #2 (Remarks to the Author):

In this paper, the authors first synthesized pillar[5]arenes and pillar[6]arenes and used them to quantitate 1-MNA in vitro. The authors then generated NNMT deficient mice using CRISPR/Cas9 and demonstrated that pillar[6]arenes can be used to measure 1-MNA concentration in urine.

Upon the editor’s request, the reviewer mainly evaluates the animal experiment part.

Reply: We greatly appreciate reviewer 2’s comments about the animal and biological experiments.

1. The animal experiments are appropriately performed. The reviewer recommends authors to disclose the approval number form the Institutional Animal Care and Use Committee (line 415-6).

Reply: Thank you for your ethically important comments. We have provided the approval number, “AP-204142”, of our animal experiments in the “Methods” part of the revised manuscript. We have also disclosed the name of the board and institution that approved the study protocol of the animal experiments in this study.

2. The authors measured 1-MNA concentration in serum and showed it is about 1,000 times less than in urine (Fig s7). The reviewer would like to know if the P6A can be used in such a low concentration range.

Reply: Blood tests are one of the most common medical tests for diagnosing any condition. Thus, it is of interest to some readers to know whether P6A is also useful for detection of 1-MNA in plasma. However, 1-MNA could not be detected in blood plasma by P6A because the sensitivity of the method was not high enough (see the Figure below). This figure is provided in the revised “Supplementary Information (Supplementary Figure 15)”. We also discussed how to improve the sensitivity of the biosensor in the “Discussion” part of the revised manuscript.

3. Since *Nnmt1* KO samples are absolute negative control, the authors should dilute 1-MNA in KO mouse urine and use for PA6 validation.

Reply: To validate the P6A specificity to urinary 1-MNA, we diluted 1-MNA in *Nnmt* KO mouse-derived urine, and fluorescence measurements were performed (see below). We included these results in the manuscript and believe that they help to emphasise our conclusion. Thanks to the referee for this valuable suggestion.

Revision in the “Results” section: We have inserted sentences, below.

Furthermore, to validate the specificity of P6A to 1-MNA, we diluted 1-MNA in *Nnmt* KO mouse-derived urine, and fluorescence measurements were performed (Supplementary Figure 14, and Figure 6d, dotted line). The correlation obtained for the concentration of 1-MNA determined by LC-MS/MS and the values of the % inhibition obtained from the fluorescence measurements of 1-MNA diluted in *Nnmt* KO urine were similar to the results above. These data clearly indicated that P6A can specifically and quantitatively detect 1-MNA in crude biological samples.

Minor

4. The authors removed exon 2 that cause frameshift mutation. The reviewer agrees that the loss of exon2 disrupts gene function, but unexpected alternative splicing variants sometimes appear. The authors should show there are no alternative splicing variants nor functional NNMT proteins in KO mice.

Reply: We performed additional experiments and text was added in the “Results” and “Methods” sections (see below). We have also added Figures in the “Supplementary Information”. We believe that these additions help to emphasise our conclusion that our established *Nnmt* KO mouse has no functional Nnmt protein. Thank you to the referee for these suggestions.

Revision in the “Results” section:

In the *Nnmt* KO mouse, no unexpected alternative splicing variants of *Nnmt* mRNA nor functional Nnmt protein were detected in the liver (Supplementary Note 1, and Supplementary Figures 9, 10, 11, 12, and 13).

Revision in the “Supplementary Information”: We have inserted the sentences as shown below.

Supplementary Note 1

Validation of functional Nnmt deficiency in the *Nnmt* KO mouse

To verify that the *Nnmt* KO mouse had no mRNA that coded for functional Nnmt protein, we performed reverse transcription (RT)-PCR. Total RNA was purified from liver and transcribed into cDNA, and then cDNAs encoding Nnmt protein were amplified by PCR. PCR primers (Nnmt Fw1 and Rev1) were designed upstream and downstream of exon 2 to understand the structure of the exon boundaries. Agarose gel electrophoresis indicated that one major (#1 in wild-type (Wt), and #3 in *Nnmt* KO mouse) and one minor (#2 in Wt, and #4 in *Nnmt* KO mouse) transcript were detected (Supplementary Figure 9a). Each PCR product (from #1 to #4) was purified by gel extraction and individual DNA sequences were determined by subcloning. The sequences of the major (#1) and minor (#2) PCR products completely corresponded to the annotated mouse *Nnmt*-001 and 002 in the public database (Vertebrate Genome Annotation (VEGA) database), respectively (Supplementary Figures 9b and 10). In the *Nnmt* KO mouse, exon 1 of both cDNAs (#3 and #4) was directly connected to exon 3 without any insertion or deletion that would have caused a frameshift in the open reading frame of the Nnmt protein. We also designed other primer pairs (Nnmt Fw2 and Rev2) for amplification of the whole coding sequence of Nnmt, and performed RT-PCR. The cDNA sequences of the PCR products were directly determined without subcloning. In the electropherogram, we found that only a single peak per base was detected for almost all single nucleotides in the whole exon 2 in the Wt sequence and around the junction between exon 1 and 3 in the *Nnmt* KO sequence. These data suggests that no unexpected alternative splicing variant was transcribed in the *Nnmt* KO mouse (Supplementary Figure 11 and 12). Furthermore, while *Nnmt* is highly expressed in liver, no Nnmt protein was detected in the liver of the *Nnmt* KO mouse from the western blot analysis (Supplementary Figure 13). These data strongly demonstrate that the *Nnmt* KO mouse has no functional Nnmt protein.

Revision in the “Methods” section: We have inserted the sentences as indicated below.

In the public VEGA database (<http://vega.archive.ensembl.org/index.html>), five alternative splicing variants of *Nnmt* mRNA were found. Based on the Human And Vertebrate Analysis and Annotation (HAVANA), two splicing variants of *Nnmt* mRNA (*Nnmt*-001 and -002) have protein coding sequences. While the *Nnmt*-001 codes the full protein coding sequence of *Nnmt* protein and *Nnmt*-002 lacks a large part of exon 3, both mRNAs contain the whole sequence of exon 2.

5. While line 271 says “various doses of nicotinamide”, there is only one column presented without any information in figure 6.

Reply: In the results shown in Figure 6a-c, the experiments were performed using a drinking water concentration of nicotinamide of 2 mg/mL, and in the results shown in Figure 6d, the mice were treated with various concentrations of nicotinamide in water. Therefore, we have added this information in the Figure legend (see below). We have also modified Figure 6d (see attached Figure below). We hope that these revisions make the Figure more informative, and will allow readers to more easily understand that intake of nicotinamide dose-dependently increases urinary 1-MNA excretion. We thank the referee for the important comment.

Revision in the “Figure 6 legend” section:

(a-c) In the nicotinamide treatment group (Wt + NA), the drinking water contained 2 mg/mL of nicotinamide.

(d) The linear correlation between the concentration of 1-MNA obtained from LC-MS/MS and the % inhibition obtained from the fluorescence measurements. Line represents the linear regression calculation ($n = 19$). Wild-type mice were treated with drinking water containing various doses of nicotinamide (from 0 to 2 mg/mL). The dotted line is derived from Supplementary Figure 14.

Reviewer #3 (Remarks to the Author):

I quite like the idea of this manuscript, which is to use pillar[n]arenes to target 1-methylnicotinamide. Despite that enthusiasm, there are a number of relatively minor issues that reduce my overall enthusiasm for the manuscript and will need to be addressed before I can recommend publication. These include:

1. The abstract begins with information about metabolic syndrome and the associated health risks. While this is interesting information, it is only tangentially related to the main point of the manuscript, which focuses on supramolecular complexation inside pillar[n]arene guests. As such, this information should be eliminated from the abstract and moved to the introduction section exclusively.

Reply: We thank the referee for the comments about the “Abstract”. We agree with reviewer 3’s comment and have eliminated information about metabolic syndrome from

the abstract. This revision has made our abstract more informative, and helps us to emphasise our important findings.

2. Moreover, when the abstract indicates that P6A can be used as a “turn-off” sensor, the authors should provide quantitative information about how effective this sensor is.

Reply: In the revised manuscript, we have added the information about the detection limit for 1-MNA by P6A.

3. The abstract also uses the term “nanosensor” without defining what this means or why P6A would qualify.

Reply: We agree that the term “nanosensor” is not suitable in our manuscript because some readers may associate “nanosensor” with a nano-size device. Because we claim that P6A specifically detects biological metabolites, we thought that “biosensor” is more suitable than “nanosensor”. In the revised manuscript, we have used the word “biosensor” instead of “nanosensor”.

4. In the introduction, the information about antigen-antibody binding needs to have a supporting reference.

Reply: We thank the reviewer for this correction. We have added a suitable reference about antigen-antibody binding (“Introduction”, and reference 5 and 6 in the revised manuscript).

5. Since pillar[n]arenes have been reported since 2008, it is not clear that the adjective “new” should be used for their description, as the authors do in the introduction.

Reply: In fact, since pillararenes were first reported in 2008, they have been extensively investigated in not only the supramolecular chemistry field, but also in the materials science and biomedical areas because of their ease of synthesis and versatile chemical modifiability. Therefore, the authors agree that pillararenes are not so “new” in terms of current chemistry, and we have omitted this adjective, “new”, from the sentence.

6. Also in the introduction, the authors claim that the cavity size of pillar[5]arene and pillar[6]arene are 4.7 and 6.7 angstroms. I assume the authors are referring to diameter across the cavity; if so, they should indicate that more explicitly.

Reply: We agree that we would like to refer to the diameter across the cavity. Therefore, we have used the phrase “diameter across the cavity” instead of “cavity size” in the

revised manuscript. We also have modified “Figure 1c” (see below). We included inscribed circles to indicate the diameter across the cavity.

7. In the introduction, when the authors refer to the use of “NMR,” they should be more explicit and refer to “NMR spectroscopy” and in particular differentiate between proton and carbon NMR.

Reply: Thanks to the referee for this correction, we have added the words, “¹H nuclear magnetic resonance”.

8. The authors claim that paraquat is “similar in structure” to 1-MNA. They should clarify by what metrics they are assessing this similarity.

Reply: The cationic structure of 1-MNA is similar to dimethyl viologen (paraquat). Our group and others have shown that P5A and P6A form stable 1:1 host–guest complexes with paraquat with extremely high association constants of $8.2 \pm 1.7 \times 10^4 \text{ M}^{-1}$ and $1.02 \pm 0.1 \times 10^8 \text{ M}^{-1}$, respectively. Thus, we hypothesized that the two water-soluble pillar[n]arenes P5A and P6A would be good candidates as biosensors for detection of 1-MNA (Fig. 1b). We mentioned this point in the revision as follows:

Revision in the “Results” section:

The cationic structure of 1-MNA is similar to dimethyl viologen (paraquat). Our group and others have shown that anionic P5A and P6A formed stable 1:1 host–guest complexes with paraquat with extremely high association constants of $8.2 \pm 1.7 \times 10^4 \text{ M}^{-1}$ and $1.02 \pm 0.1 \times 10^8 \text{ M}^{-1}$, respectively^{27, 28}. Thus, we hypothesised that the two water-soluble pillar[n]arenes P5A and P6A would be good candidates as biosensors for detection of 1-MNA (Fig. 1b).

9. I am concerned about the lack of control experiments reported herein. The authors claim that the reason for selectivity of the pillar[n]arene is related to the size of the

cavity, but do not exclude that ionic interactions may be solely responsible for the observed effects. More information should be provided to explain how the authors are accounting for favorable host-guest electrostatic interactions.

Reply: As the reviewer mentioned, we also think electrostatic interactions between the pyridinium cation of 1-MNA and the carboxylate anions of P6A are involved in forming the host-guest complex. Actually, 2py is a similar size to 1-MNA, but is not a cation; thus, P6A has a lower binding affinity to 2py compared with 1-MNA. These data clearly indicated that electrostatic interactions between the pyridinium cation of 1-MNA and carboxylate anions of P6A also contribute to forming a host-guest complex between P6A and 1-MNA. We mentioned these points in the revised manuscript as follows:

Revision in the “Results” section:

2py is a similar size to 1-MNA, but is not a cation; thus, P6A has a lower binding affinity to 2py compared with 1-MNA. These data clearly indicated that electrostatic interactions between the pyridinium cation of 1-MNA and carboxylate anions of P6A were also involved in the formation of a host-guest complex between P6A and 1-MNA.

10. The authors claim that photoinduced electron transfer is occurring between the guest and the host. It is far from clear that photoinduced electron transfer is in fact the mechanism by which such fluorescence decreases are observed. The authors should justify the proposed mechanism and/or provide relevant supporting references.

According to the reviewer’s comment, we provided relevant supporting references as follows:

Papers mentioned the photoinduced electron transfer of pillar[n]arenes:

Chem. Commun. 2010, 46, 3708–3710, *J. Am. Chem. Soc.* 2012, 134, 19489–19497.

Also we added three review papers about photoinduced electron transfer using supramolecular systems as follows:

Chem. Soc. Rev. 2015,44, 4203–4211, *Acc. Chem. Res.* 2019, 52, 10, 2818–2831, *Chem. Soc. Rev.* 2015,44, 4192–4202.

11. In the section focused on detection limits, the authors should provide information about other systems that detect these metabolites and what the limits of detection for these systems are.

Reply: The detection limit for 1-MNA is <50 nM by the LC-MS/MS method. We have added a sentence in the “Discussion” section (see Reply to #12 below).

12. The fact that the system works in unpurified urine is certainly interesting, but urine is actually a fairly simple biological fluid. Have the authors conducted any testing in more complex biological systems? If so, they should include results here. If not, they should consider doing so and/or at a minimum provide a hypothesis as to how well the system would work.

Reply: We also tried to detect 1-MNA in blood samples. However, 1-MNA in blood plasma was not detected using P6A because the method was not sensitive enough (see Figure below). This figure is provided in the revised Supplementary Information. We also discussed the possibility of how to improve the sensitivity of the biosensor in the “Discussion” part of revised manuscript.

Revision in the “Discussion” section: We have inserted the sentences as shown below.

In this research, we found that P6A can be used to specifically detect 1-MNA in urine. However, compared with the LC-MS/MS method (detection limit is <50 nM), the sensitivity of P6A to 1-MNA was very low, therefore, 1-MNA in blood plasma could not be detected using P6A (Supplementary Figure 15). To improve the sensitivity of P6A for 1-MNA, further chemical modification is required. One significant and common problem in the detection of specific molecules by fluorescence biosensors is “autofluorescence”, that is, fluorescence from inherent components of biological samples. To increase the fluorescent signal of the P6A, surface modification by pillararene may be useful to develop highly-sensitive sensor materials. Previously, we reported that immobilized multilayer pillar[5]arene films could efficiently absorb a guest molecule, para-dinitrobenzene (*p*-DNB), and the adsorbed amount of *p*-DNB exponentially increases with an increasing number of pillar[5]arene-layers⁴⁶. Further experiments will help to improve the sensitivity and specificity of the biosensors, and ultimately, this work should contribute to the development of low-cost, easy, and rapid methods for the detection of human metabolites for diagnosis.

Revised “Supplementary Information” contains the Figure shown below.

Sincerely yours,

T. Ogoshi

Tomoki Ogoshi

Dr. Tomoki Ogoshi
Department of Synthetic Chemistry and Biological Chemistry,
Graduate School of Engineering
Kyoto University, Katsura, Nishikyo-ku, Kyoto, 615-8510 Japan
(E-mail) ogoshi@sbchem.kyoto-u.ac.jp
(Phone) +81(Japan)-75-383-2733
(FAX) +81(Japan)-75-383-2732

REVIEWERS' COMMENTS:

Reviewer #1 (Remarks to the Author):

I am glad to see that the quality of the article is very good. This manuscript is now acceptable for publication in its current form.

Reviewer #2 (Remarks to the Author):

The authors satisfactory addressed my concerns.

Reviewer #3 (Remarks to the Author):

The authors have done an outstanding job responding to my comments on the previous version of the manuscript. I am pleased to recommend that the manuscript be accepted for submission in its current state.